# Decay Resistance of Nano-Zinc Oxide, and PEG 6000, and Thermally Modified Wood

**Ladislav Reinprecht \*, Miroslav Repák, Ján Iždinský and Zuzana Vidholdová**

Department of Wood Technology, Faculty of Wood Sciences and Technology, Technical University in Zvolen, T. G. Masaryka 24, 96001 Zvolen, Slovakia; miroslav.repak@pobox.sk (M.R.); jan.izdinsky@tuzvo.sk (J.I.); zuzana.vidholdova@tuzvo.sk (Z.V.)

\* Correspondence: reinprecht@tuzvo.sk

**Abstract:** In Central Europe, European beech (*Fagus sylvatica* L.) wood has a high potential for the production of construction and decorative materials, with the aim of replacing Norway spruce, oaks, and other traditionally used tree species. However, the biological resistance of beech wood—to decaying fungi, molds, and insects—is low, and in damp conditions its resistance must be increased with suitable preservatives or modification methods. In the present experiment, beech wood was first treated with water systems of nano-zinc oxide (0.1 to 3.3 wt.% of nano-ZnO) and/or polyethylene glycol 6000 (20 wt.% of PEG 6000), without/with additional thermal modification at 190 °C/2 h. In the presence of nano-ZnO, the decay resistance of beech wood to the brown-rot fungus *Rhodonia placenta* and the white-rot fungus *Trametes versicolor* significantly increased, mainly after its additional thermal modification. The presence of nano-ZnO in beech wood—(a) alone, (b) with a subsequent application of PEG 6000, (c) with additional thermal modification—had a more apparent inhibition effect on *T. versicolor* than on *R. placenta*. PEG 6000 alone did not improve the resistance of beech wood to rot.

**Keywords:** beech wood; nano-zinc oxide; polyethylene glycol; thermal modification; rot fungi; *Rhodonia placenta*; *Trametes versicolor*

## 1. Introduction

The European beech (*Fagus sylvatica* L.) belongs to the most commercially used hardwoods in Central Europe, due to its significant reserves in forests. In the Slovak forests, deciduous trees predominate at 63.9%, with mainly beech (34.6%) and oak (10.4%) among them [1]. In general, beech wood is classified as being a medium–high density hardwood with good workability [2]. Beech wood is used as sawn timber mainly in furniture production, e.g., for upholstered furniture construction frames or after hydrothermal treatment for bent seating furniture elements. As glued laminated timber, it is used for the production of stairs, floors, prisms for railways, and industrial sleepers. Veneers for decorative purposes for furniture production and interior purposes are produced from the highest quality parts of beech wood. Flat and shaped boards are made of plywood from technical beech veneers. A large amount of beech wood is also used to make boxes, toys, and pallets, and also as fuel wood [3]. However, all parts of beech wood, i.e., its sapwood, heartwood (also called mature-wood), and red-false heartwood, have several disadvantages—mainly a lower dimensional stability in a humid environment, and a low resistance to decay by fungi and damages by insects [4]. A low biological resistance of beech wood to damaging fungi and insects has to be overcome in specific situations in interiors (e.g., in bathrooms or kitchens, and also for walls, ceilings, and roofs with the possibility of water vapor condensation) and commonly in exteriors (e.g., prisms for sleepers and pilots); this means mainly that it is used in higher usage classes from 2 to 5 by EN 335 [5], with suitable thermal and chemical modification methods, or

with environmentally acceptable biocides.

Nanoproducts that are used to protect wood against biodegradation and weathering are divided into two types: (I) Nanocapsule refers to a biocide that is embedded in a polymeric nanocarrier, while (II) nanomaterial is nanosized metal that can be directly impregnated into wood or added to coatings [6,7]. Nanosized metal oxides, such as zinc oxide (nano-ZnO), titanium dioxide (nano-TiO$_2$), and cerium dioxide (nano-CeO$_2$) have strong antimicrobial properties [8]. Due to their very small size, nano-ZnO and other nanosized metals can penetrate deeply and uniformly into the wood pores, leading to homogenous protection of the wood substrate while reducing the issue of excessive leaching. Nanoparticles increase the decay resistance of wood by reducing the moisture availability in the wood, either by preventing the absorption of the moisture, or by blocking the flow path of liquid water. More researchers—e.g., [9–16], have reported that nano-ZnO, when used for wood protection, is very effective against white-rot fungi; it partially inhibits the activity of brown-rot fungi, while its efficiency against molds is poorer. Bak and Németh [17] studied the efficacy of different nanoparticles, namely zinc oxide, zinc borate, copper borate, silver, and copper, against fungi. They found that the most effective nanoparticle treatments were those containing zinc oxide and borate (zinc borate and copper borate). However, only zinc oxide provided effective protection after leaching.

Polyethylene glycols are polar liquids (lower molecular PEG 300 to PEG 1000) or solids with a waxy consistency (higher molecular PEG 1500 to PEG 10,000). PEGs are successfully used for the dimensional stabilization of waterlogged wooden archaeological artefacts [18–20]. However, the lower molecular PEGs are not able to increase the strength of wood, because wood that is treated with them has permanently swollen cell walls, and the strengthening effect of hydrogen bonds inside the cell walls, which consist of polysaccharides and lignin, is suppressed [21].

A classic thermal modification process for wood in hot air is usually performed at optimal temperatures from 180 to 220 °C, when significant changes occur in its molecular structure—especially due to the degradation of certain parts of hemicelluloses, the extinction of some polar hydroxyl groups, and the creation of new 3D hemicellulose–lignin linkages—while only slight changes occur in their anatomical and geometric structure [22–27]. Structural changes in the thermally modified wood are reflected in its increased dimensional stability and biological resistance, while its strength is partially decreased [28–30]. With the aim of improving specific properties of wood products, a combination of the thermal and chemical modifications of wood can be performed. Kamperidou [31] subjected black pine wood to thermal modification at 180 and 200 °C/3–7 h, followed by the chemical treatment of its surfaces with organosilanes. The thermally and chemically modified pine wood had better biological resistance against the brown rot fungus *Coniophora puteana*, and also against molds, when compared to unmodified and thermally modified woods. Combined PEG–thermal modification of wood at 140 °C was studied by Luo et al. [32], in order to improve the dimensional stability of wood–polypropylene (WPP) composites. Scanning electron microscope analysis indicated that the primary "bulking" effect of PEG on wood particles, which resulted in the reduction of water uptake by WPP composites, had a negative effect on their flexural modulus of rupture and modulus of elasticity.

The aim of this work was based on an assumption that combining a primary treatment of beech wood with an environmentally acceptable biocide, nano-ZnO, and its subsequent thermal modification could more effectively increase the biological resistance of this nondurable tree species to decaying fungi. At the same time, the anti-decay effect of nano-ZnO in combination with PEG 6000 and thermal modification was monitored.

## 2. Materials and Methods

### 2.1. Beech Wood

In total, 342 European beech (*Fagus sylvatica* L.) heartwood specimens 25 mm × 25 mm × 5 mm (longitudinal × tangential × radial) of high quality (228 for modification and 114 references), were used in the experiment; i.e., specimens without rot, insect galleries, growth defects, tension wood, and red-false wood. Specimens were sawn from naturally seasoned boards prepared from one 82-year-old beech tree trunk obtained from the National Forest Centrum in Zvolen. Specimens were then dried at 103 ± 1 °C to an oven-dry state in the Memmert UNB 100 kiln (Memmert GmbH + CoKG, Schwabach, Germany), and subsequently cooled in desiccators to 20 ± 2 °C, and finally weighed with an accuracy of 0.001 g ($m_0$).

### 2.2. Chemical Treatment of Beech Wood with Nano-ZnO and/or PEG 6000

The individual groups of beech wood specimens were chemically treated using immersion technology in stainless steel containers with the following compounds: (A) 0.1, 0.33, 1.0, and 3.3 wt.% water systems of nano-ZnO (Sigma-Aldrich Co., Ltd., Munich, Germany: <100 nm particle size (DLS); <35 nm average particle size (APS); 50 wt.% water solution), and (B) 20 wt.% water solution of PEG 6000 (HiMedia, Laboratories Pvt. Ltd., Mumbia, India: solid waxy substance; a prepared 20 wt.% water solution with a dynamic viscosity of 21.8 × 10$^{-3}$ Pa·s at 20 °C). All chemical treatments were performed at atmospheric pressure for 3 h, either at 20 ± 2 °C, using water systems of nano-ZnO, or at 100 ± 2 °C, using a water solution of PEG 6000. Chemically treated specimens were then conditioned for 14 days to a moisture content of approximately 10% at a temperature of 20 ± 2 °C and a relative air humidity of 60% ± 3%, and then dried at 103 ± 1 °C to an oven-dry state, cooled in desiccators to 20 ± 2 °C, and finally weighed with an accuracy of 0.001 g ($m_{0/Chemically\text{-}Treated/}$). The weight percentage gain (WPG) of nano-ZnO or PEG 6000 was determined using Equation (1):

$$WPG = \frac{m_{0/Chemically-Treated/} - m_0}{m_0} \cdot 100 \ (\%), \tag{1}$$

where $m_{0/Chemically\text{-}Treated/}$ is the mass of the oven-dried chemically treated specimen containing nano-ZnO or PEG 6000 (g), and $m_0$ is the mass of the original oven-dried specimen (g).

### 2.3. Thermal Modification of Beech Wood

Thermal modification (TM) of the reference and chemically treated beech wood specimens was performed in a Memmert UNB 100 kiln (Memmert GmbH + CoKG, Schwabach, Germany) at a temperature of 190 °C for 2 h. This mode of the TM process was chosen on the basis of results obtained in our previous experiments [33,34]. Specimens were then cooled in desiccators to 20 ± 2 °C, weighed with an accuracy of 0.001 g ($m_{0/Thermally\text{-}Modified/}$, or $m_{0/Chemically\text{-}Thermally\text{-}Modified/}$), and their mass losses due to the TM process were evaluated as percentages.

### 2.4. Resistance of Beech Wood to Decaying Fungi

Fungal attacks of the reference—original, chemically treated, thermally modified, and chemically and thermally modified beech wood specimens were performed with the brown-rot fungus *Rhodonia placenta* (Fr.) Niemelä, K.H. Larss. & Schigel (synonyms *Poria placenta* (Fries) Cooke sensu J. Eriksson or *Postia placenta* (Fries) M.J. Larsen & Lombard), strain FPRL 280 (Building Research Establishment, Garston-Watford-Herst, Watford, UK) and the white-rot fungus *Trametes versicolor* (Linnaeus ex Fries) Pilat, strain BAM 116 (Bundesanstalt für Materialforschung und -prüfung, Berlin, Germany). Fungal attacks took place in accordance with testing for the fungal resistance of beech wood that had been thermally modified in the melt of paraffin [33], or in the melt of PEG 6000 [34], and similarly to the rapid mycological screening test by Van Acker et al. [35]; i.e., in con-

trast to the standard EN 113-2 [36]—using smaller dimensions of specimens, their other sterilization method, and a shorter incubation time.

In the vaccination box (Merci, Ferrara, Italy), two specimens of the equally treated type and one reference control specimen were placed into a Petri dish with a diameter of 100 mm on plastic mats under which a fungal mycelium was already grown up on a 3–4 mm thick layer of 4.5% malt agar soil (HiMedia, Laboratories Pvt. Ltd., Mumbia, India). The incubation process took 6 weeks at a temperature of 24 ± 2 °C and a relative air humidity of 90% ± 5%. After the decay test, the specimens were dried at 103 ± 1 °C to an oven-dry state, cooled in desiccators to 20 ± 2 °C, and finally, weighed with an accuracy of 0.001 g ($m_{/After\text{-}Fungal\text{-}Attack/}$) to determine their mass losses (Δm) using Equation (2):

$$\Delta m = \frac{m_{/Before-Fungal-Attack/} - m_{/After-Fungal-Attack/}}{m_{/Before-Fungal-Attack/}} \cdot 100 \ (\%), \tag{2}$$

where $m_{/Before\text{-}Fungal\text{-}Attack/}$ ($m_0$, $m_{0/Chemically\text{-}Treated/}$, $m_{0/Thermally\text{-}Modified/}$, or $m_{0/Chemically\text{-}Thermally\text{-}Modified/}$) is the mass of the specimen in the oven-dry state before the fungal attack (g), and $m_{/After\text{-}Fungal\text{-}Attack/}$ is the mass of specimen in the oven-dry state after the fungal attack (g).

## 3. Results and Discussion

### 3.1. Weight Percentage Gain of Chemicals

The WPG values of nano-ZnO in the beech wood specimens ranged from 0.047% to 1.548%, and these values corresponded well, with a concentration increase in ZnO nanoparticles in water systems from 0.1% to 3.3% (Table 1). The WPG values of PEG 6000 ranged over a narrow interval from 8.14% to 9.72%, and this result indirectly means that they were not apparently influenced by the potentially non-homogenous permeability of the individual beech wood specimens, or by their previous treatment with nano-ZnO (Table 1).

**Table 1.** Weight percentage gains (WPGs) of nano-ZnO and PEG 6000 into beech wood specimens.

| Treatment of Beech Wood | Nano-ZnO WPG (%) | PEG 6000 WPG (%) |
|---|---|---|
| ZnO 0.1% | 0.051 | - |
| ZnO 0.33% | 0.148 | - |
| ZnO 1.0% | 0.532 | - |
| ZnO 3.3% | 1.548 | - |
| PEG 6000 20% | - | 8.36 |
| ZnO 0.1% + PEG 6000 20% | 0.047 | 9.18 |
| ZnO 0.33% + PEG 6000 20% | 0.161 | 8.14 |
| ZnO 1.0% + PEG 6000 20% | 0.568 | 9.64 |
| ZnO 3.3% + PEG 6000 20% | 1.497 | 9.72 |

Note: The average values were determined from 24 specimens.

### 3.2. Mass Loss at Thermal Modification

The mass losses of the beech wood specimens caused by their thermal modification at 190 °C over 2 h ($\Delta m_{Thermally\text{-}Modified}$) ranged from 3.65% to 4.72%, with an average value of 4.18%.

### 3.3. Decay Resistance

The original reference beech wood specimens proved to be non-sufficiently resistant to rot caused by wood-decaying fungi, whereas their mass losses in mycological tests were more than 20%, i.e., 22.73% by *Rhodonia placenta* and 21.81% by *Trametes versicolor* (Table 2, Figures 1–3). PEG 6000 did not increase the decay resistance of beech wood. Similarly, other studies with synthetic polymeric stabilizers and consolidants,

such as polyvinylchloride, polyurethane, or polyacrylates, have also shown no or poor efficiency with regard to the colonization and degradation of wood by fungi [13,37–41]. On the contrary, nano-ZnO evidently increased the decay resistance of beech wood. This was documented by significant and exponentially decreased mass losses ($\Delta m$) of beech specimens caused by *R. placenta* and *T. versicolor* with a higher concentration of nano-ZnO ($C_{ZnO}$) when used alone, as well as in combinations with PEG 6000, TM, or PEG 6000 + TM (Table 2, Figures 1 and 2).

With nano-ZnO only, with an increase in concentration from 0% to 3.3%, the decay of beech wood by fungal attack significantly decreased when the coefficients of determination of exponential relations, $R^2$, were 0.892 for *R. placenta* and 0.48 for *T. versicolor* (Table 2, Figures 1 and 2). This is in accordance with the research in [14,42,43]. Within the chemical treatments of beech wood, the maximum increase in its decay resistance was observed when using the highest concentration of nano-ZnO (3.3%), i.e., against *T. versicolor* by 96.1% ($\Delta m$ from 21.81% to 0.86%), and against *R. placenta* by 76.9% ($\Delta m$ from 22.73% to 5.25%), (Table 2, Figures 1 and 2).

**Table 2.** Mass losses ($\Delta m$) of reference, chemically treated, thermally modified, and chemically-thermally modified beech wood specimens in mycological tests caused by the wood-decaying fungi *Rhodonia placenta* and *Trametes versicolor*.

| Treatment of Beech Wood | *R. placenta* $\Delta m$ (%) | *T. versicolor* $\Delta m$ (%) |
|---|---|---|
| Original–Reference | 22.73 (2.86) | 21.81 (4.02) |
| ZnO 0.1% | 19.27 (4.16) d | 15.19 (2.11) c |
| ZnO 0.33% | 16.68 (2.46) b | 4.25 (2.26) a |
| ZnO 1.0% | 12.29 (2.42) a | 3.88 (1.82) a |
| ZnO 3.3% | 5.25 (1.98) a | 0.86 (0.06) a |
| PEG 6000 20% | 24.40 (2.47) d | 21.96 (2.76) d |
| ZnO 0.1% + PEG 6000 20% | 20.32 (4.52) d | 19.98 (2.08) d |
| ZnO 0.33% + PEG 6000 20% | 20.14 (3.14) d | 11.14 (2.74) a |
| ZnO 1.0% + PEG 6000 20% | 14.54 (2.25) a | 5.47 (0.47) a |
| ZnO 3.3% + PEG 6000 20% | 10.13 (4.49) a | 4.65 (0.36) a |
| TM 190 °C/2 h | 18.97 (2.22) c | 10.50 (1.08) a |
| ZnO 0.1% + TM 190 °C/2 h | 17.66 (2.76) b | 11.27 (1.28) a |
| ZnO 0.33% + TM 190 °C/2 h | 14.01 (1.49) a | 8.20 (1.61) a |
| ZnO 1.0% + TM 190 °C/2 h | 11.20 (4.49) a | 3.74 (0.71) a |
| ZnO 3.3% + TM 190 °C/2 h | 2.63 (2.13) a | 0.18 (0.14) a |
| PEG 6000 20% + TM 190 °C/2 h | 15.75 (2.99) a | 8.63 (1.76) a |
| ZnO 0.1% + PEG 6000 20% + TM 190 °C/2 h | 15.62 (2.41) a | 5.39 (1.38) a |
| ZnO 0.33% + PEG 6000 20% + TM 190 °C/2 h | 9.38 (3.65) a | 6.03 (1.28) a |
| ZnO 1.0% + PEG 6000 20% + TM 190 °C/2 h | 5.23 (2.83) a | 2.38 (0.18) a |
| ZnO 3.3% + PEG 6000 20% + TM 190 °C/2 h | 2.29 (0.37) a | 1.73 (0.46) a |

Note: The average values of modified beech wood series (19 series in total) were determined from 6 specimens, and reference series from 57 specimens. Standard deviations are given in parentheses. The Duncan tests were performed in relation to the reference beech wood specimens, and their evaluation was performed at the levels of significance a = 99.9%, b = 99%, c = 95%, and d < 95%.

In the combined nano-ZnO and thermal modifications of beech wood, the anti-decay effect of nano-ZnO with its usage at higher concentrations was also significant with exponential course, achieving a high $R^2$ of 0.946 for *R. placenta* and 0.853 for *T. versicolor*. The summary or even synergistic effect of the two modification methods—chemical and thermal—is probably manifested here. The best and, for practical purposes, acceptable high decay resistance was determined in the case of the beech wood spec-

imens that were initially treated with a 3.3% aqueous system of nano-ZnO and subsequently subjected to thermal modification at a temperature of 190 °C/2 h—with an average mass loss of below 3% (Table 2, Figures 1 and 2). In addition, nano-ZnO has a strong interaction with the chemical constituents of wood such as hemicelluloses and lignin, and it can subsequently reduce the water absorption and volumetric swelling of wood [44].

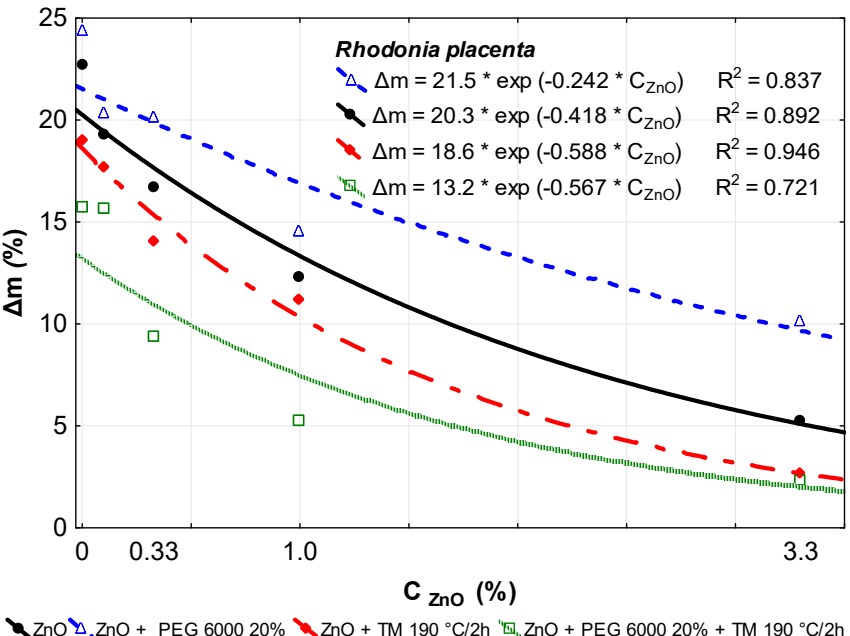

**Figure 1.** Exponential correlations between the concentrations of nano-ZnO (C$_{ZnO}$) used for the chemical and chemical–thermal modifications of beech wood and the decay attack of modified beech wood specimens with the brown-rot fungus *R. placenta*, evaluated on the basis of their mass losses (Δm).

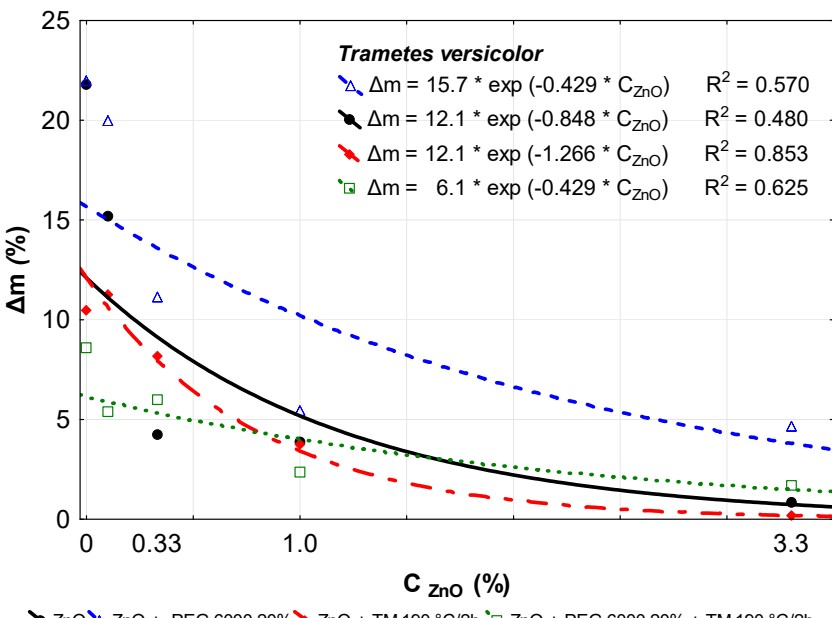

**Figure 2.** Exponential correlations between the concentrations of nano-ZnO (C$_{ZnO}$) used for the chemical and chemical–thermal modifications of beech wood and the decay attack of modified beech wood specimens with the white-rot fungus *T. versicolor*, evaluated on the basis of their mass losses (Δm).

For all treatments of beech wood with nano-ZnO—(a) used alone, (b) used with the following application of PEG 6000, (c) used with subsequent thermal modification—this inorganic fungicide appeared to be more effective against *T. versicolor* compared to *R. placenta* (Table 2, Figures 1 and 2). This knowledge was confirmed using the Duncan tests as well, when for *T. versicolor* the highest significance levels of 99.9% of the decay difference resistance between the reference and modified specimens were already achieved at relatively lower concentrations of nano-ZnO (Table 2). From several literature sources, it is well known, in accordance with the results of our experiment, that fungicides based on zinc and other heavy metals are more efficient against white-rot fungi compared to brown-rot fungi [9,10,12,15,45]. For example, in Clausen et al. [9], poplar and pine sapwoods treated with 1%, 2.5%, and 5% water solutions of nano-ZnO were observed to have an evidently higher decay resistance against the white-rot fungus *T. versicolor* compared to nonsufficient decay resistance against the brown-rot fungi *Rhodonia (Postia) placenta* and *Gloeophyllum trabeum*. However, from Németh et al. [46], nano-ZnO significantly increased the decay resistance of beech to the brown-rot fungus *Rhodonia (Poria) placenta*, as well.

**Figure 3.** Examples from the mycelia growth of the wood-decaying fungi (**a**) *R. placenta* and (**b**) *T. versicolor* on the surfaces of differently modified beech woods. Note: In each Petri dish there are two modified specimens (up) and one reference specimen (down).

The thermal modification of beech wood itself also had a positive effect on its decay resistance, which was more apparent with *T. versicolor*, at 51.9% ($\Delta$m from 21.81% to 10.50%), compared to *R. placenta*, by 16.5% (Table 2). The combined chemical–thermal modification of beech wood in the presence of PEG, i.e., firstly in a 20% aqueous solution of PEG 6000, followed by 190 °C/2 h, was only partially more effective—the resistance of such modified beech wood to *T. versicolor* increased maximally by 60.4%, and to *R. placenta*, it increased maximally by 30.7% (Table 2). However, the additional thermal modification process apparently increased the decay resistance of the beech wood specimens

that were first chemically treated with nano-ZnO, i.e., against *T. versicolor* by a maximum of 99.2% (Δm from 21.84% to 0.18%), and against *R. placenta* by a maximum of 88.4% (Δm from 22.73% to 2.63%), (Table 2, Figures 1 and 2). Study [44] reported that nano-treatment and increased temperature clearly reduced the number of hydroxyl groups, affected hemicelluloses, and partially deformed cellulose and lignin in wood. These changes in the chemical composition of the treated wood led to a reduction in the maximum moisture capacity of the cell walls, and a consequent increase in the resistance of the treated wood to fungal decomposition.

On contrary, the combination of two chemical treatment types, i.e., beech wood that was first treated with bio-active nano-ZnO and then with bio-inert PEG 6000, was usually less effective and resulted in a weakening of the beech wood decay resistance obtained due to nano-ZnO. This is in line with the work of Reinprecht et al. [13] in which a reduction in the fungicidal effectiveness of nano-ZnO was attributed to the "hypothesis" of a steric blockade of ZnO nanoparticles by macromolecules of acrylate resin, and thus a less frequent contact of this fungicide with wood-decaying fungi and their enzymes.

Bak et al. [47] studied the effect of nano-ZnO on increasing the decay resistance of beech and poplar woods to the brown-rot fungus *R. placenta*. Applying 0.22% nano-ZnO, at the 16-week-long mycological test, an approximately 46% lower weight loss in the chemically treated wood specimens occurred compared to the reference ones. In the presence of nano-ZnO, the increase in the decay resistance of pine wood to the brown-rot fungus *C. puteana* was only minimal [48]; however, to the other brown-rot fungus, *Serpula lacrymans*, it was already greater [11]. Bak and Németh [17] reported higher leaching resistance of nano-ZnO from wood compared to the leaching of nano-particles based on copper and borate compounds, at which nano-ZnO partially better reduced fungal activity in brown rot and more effectively inhibited it in white rot.

Marzbani et al. [10] studied the biocidal effect of nano-ZnO in particleboards (PBs). When using a 5% weight fraction of nano-ZnO, the PBs reached increased resistance to *T. versicolor* by 69.4% and to *C. puteana* by 58.4%, while when using the highest 15% weight fraction of nano-ZnO, the decay resistance of PBs to *T. versicolor* increased by 93.3% and to *C. puteana* by 81.7%. Reinprecht et al. [15] prepared PBs in which nano-ZnO was added directly into urea-melamine-formaldehyde resin in amounts ranging from 2% to 24%, at which the resistance of such prepared PBs to the brown-rot fungus *C. puteana* increased in relation to reference PBs from 33.3% to 85.7%.

### 4. Conclusions

1.  Biologically active nano-ZnO, applied as 0.1 to 3.3 wt.% water system into beech wood using an immersion techniqueof, significantly increased the decay resistance of this nondurable tree species.
2.  On the contrary, PEG 6000 did not increase the decay resistance of beech wood.
3.  Beech wood that was thermally modified at 190 °C/2 h significantly increased its resistance to the white rot fungus *Trametes versicolor* by 51.9%, but only slightly to the brown-rot fungus *Rhodonia placenta*, at 16.5%.
4.  The additional thermal modification of beech wood treated first with nano-ZnO, or with nano-ZnO and PEG 6000, further increased its decay resistance to T. *versicolor*—maximally by 99.2%, and to *R. placenta* maximally by 89.9%.
5.  Conversely, the good fungicidal efficiency of nano-ZnO was, in several cases, reduced in the case of combination with PEG 6000. This was explained by the steric-mechanical barrier of the bio-inert polyethylene glycol macromolecules—created between the biocidal active nanoparticles of ZnO, and the mycelia of the wood-decaying fungi or their enzymes.
6.  Nano-ZnO introduced into beech wood in all combinations—alone, before PEG 6000, or before thermal modification—was more effective against the white-rot fungus *T. versicolor* than against the brown-rot fungus *R. placenta*.

7. Taking into account the need for drying wood that has been chemically treated with aqueous systems of nano-ZnO and PEG, a combined chemical–thermal modification of wood, realized in one technological step, e.g., in the melt of paraffin + nano-ZnO or PEG + nano-ZnO, could be more suitable in practice, as outlined our previous experiments with thermal treatments of wood in the melts of pure paraffin or PEG [33,34].

**Author Contributions:** Conceptualization, L.R., M.R.; methodology, L.R.; software, M.R., J.I.; validation, L.R., M.R., J.I., Z.V.; formal analysis, L.R., J.I.; investigation, L.R., M.R.; resources, L.R., M.R., Z.V.; data curation, L.R., M.R., J.I.; writing—original draft preparation, L.R., M.R., J.I., Z.V.; writing—review and editing, L.R., J.I., Z.V.; visualization, L.R., J.I.; supervision, L.R.; project administration, L.R., J.I.; funding acquisition, L.R. All authors have read and agreed to the published version of the manuscript.

**Funding:** This work was supported by the Slovak Research and Development Agency under the contract No. APPV-17-0583, "Construction and decorative materials based on recycled and modified wood". Its results also came from the project implementation, "Progressive research of performance properties of wood-based materials and products (LignoPro), ITMS: 313011T720 supported by the Operational Programme Integrated Infrastructure (OPII) funded by the ERDF".

**Data Availability Statement:** Not applicable.

**Acknowledgments:** This work was supported by the Slovak Research and Development Agency under the contract No. APPV-17-0583, and by LignoPro ITMS 313011T720.

**Conflicts of Interest:** The authors declare no conflict of interest.

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
