# Peer review of "Decay Resistance of Nano-Zinc Oxide, and PEG 6000, and Thermally Modified Wood"

_forests, doi:10.3390/f13050731_

Round 1

Reviewer 1 Report

The topic of the manuscript is interesting and the paper itself provides new information. Nevertheless, there are several issues to be addressed towards the improvement of this work. In line 14, you rather use the word "primarily". In line 32, the term "glued wood" should be probably clarified (is the term Glued laminated timber more appropriate?). In line 38, do you mean the sapwood, the heartwood or both parts? Since the introduction is quite short and the state-of-the-art only roughly approached, I would propose to the authors to incorporate a published study dealing also with a treatment combination of chemical and thermal modification, the https://www.mdpi.com/1999-4907/10/12/1111 , that deals as well and focuses on the biological durability of chemically-thermally treated wood. In line 53, there is a grammatical error (missing noun). In line 73, please provide the most crucail finding of Luo et al. [31], not just referring that they worked on this treatment combination.

In materials-methods section, please refer to the following: where did the trunk was obtained from?, Did you use sapwood or heartwood or both? (since the biological durability of them differs), Where did you base the dimensions of the specimens used for the experiment?, Where did you base the specific conditions of thermal treatment? (probably a state-of-the-art description in the introduction would also touch as well briefly the optimal conditions of thermal treatment according to the literature), What was the reason why you have chosen to implement first the chemical and secondly the thermal treatment? (do you believe that the opposite order would change the results?). As regards the thermal treatment, did you record the mass loss attributed to the the wood thermal degradation? There is not any statistical analysis referred to in the text. Please clarify if there was any analysis of the results applied. In the graphs, the latin names should be presented in italics. The legends of the graphs should just explain what is presented, not providing comments on the findings. How do you explain that in the 2 last images of figure 3, concerning the Trametes versicolor, the mycelia did not totally cover the surface of the petri dish? (since as far as I know, first of all, the culture initially covers all the surface of the glass/petri and then the specimens are being placed). 

Author Response

Answers to Reviewer No 1:

The topic of the manuscript is interesting and the paper itself provides new information. Nevertheless, there are several issues to be addressed towards the improvement of this work.

In line 14, you rather use the word "primarily".

  • By us the word “primary” is better in the context of the sentence, however, now we changed it with the word “at first”.

In line 32, the term "glued wood" should be probably clarified (is the term Glued laminated timber more appropriate?).

  • Yes, we accepted this term.

In line 38, do you mean the sapwood, the heartwood or both parts?

  • We mean all parts of beech wood, because by EN 350 all its parts are easily biodegradable by decaying fungi (class 5).

Since the introduction is quite short and the state-of-the-art only roughly approached, I would propose to the authors to incorporate a published study dealing also with a treatment combination of chemical and thermal modification, the https://www.mdpi.com/1999-4907/10/12/1111, that deals as well and focuses on the biological durability of chemically-thermally treated wood.

  • Yes, the article of Kamperidou we incorporated into text of the manuscript.

In line 53, there is a grammatical error (missing noun).

  • Yes, we corrected the error, so now it is “… white-rot fungi, it partly less inhibits … “.

In line 73, please provide the most crucial finding of Luo et al. [31], not just referring that they worked on this treatment combination.

  • Yes, we now described the most important finding of these authors.

In materials-methods section, please refer to the following:

Where did the trunk was obtained from?

  • The beech trunk was obtained from the National Forest Centrum in Zvolen.

Did you use sapwood or heartwood or both? (since the biological durability of them differs).

  • We used heartwood (it is called as well as mature-wood, as beech don’t have the typical heartwood, but it can have red-false heartwood).

Where did you base the dimensions of the specimens used for the experiment?

  • On the basis of our previous thermal modifications of beech wood in the melts of paraffin and PEG (Reinprecht and Repák 2019, 2022), as well as on the basis of mycological-screening test by Van Acker et al. (2003) – see references in the chapter 2.4.

Where did you base the specific conditions of thermal treatment? (probably a state-of-the-art description in the introduction would also touch as well briefly the optimal conditions of thermal treatment according to the literature).

  • In the Introduction is now mentioned that the optimal temperatures for TM of wood are from 180 to 220 °C. In this work we used temperature of 190 °C/2h, coming out from results of our previous experiments with TM of beech in melts of paraffin and PEG at 170, 190 or 210 °C during 1, 2, 3 or 4h (Reinprecht and Repák 2019, 2022).    

What was the reason why you have chosen to implement first the chemical and secondly the thermal treatment? (do you believe that the opposite order would change the results?).

  • This order of modification techniques was chosen with the aim to better stabilize the inorganic biocide nano-ZnO in wood structure and with achieve the synergic effect of this biocide and TM. By our opinion, at using the opposite order of modification processes the results could be a partly different, because penetration of water systems of chemicals into a more hydrophobic thermally modified wood structure could be partly slowed down.   

As regards the thermal treatment, did you record the mass loss attributed to the wood thermal degradation?

  • The average mass loss of beech wood at its thermal modification or “thermal degradation” at 190 °C/2h was 4.18% (see new chapter 3.2).
  • NOTE: The mass losses of beech wood specimens caused at thermal modifications were not reflected into the mass losses caused by decaying fungi, because the initial weights of specimens before fungal attack m/Before-Fungal-Attack/ depended on their previous modification processes Þ m0, m0/Chemically-Treated/, m0/Thermally-Modified/, or m0/Chemically-Thermally-Modified/) – See Eq. 2.

There is not any statistical analysis referred to in the text. Please clarify if there was any analysis of the results applied.

  • Yes, now results of the Duncan tests and Exponential correlations were incorporated into the text of manuscript.

In the graphs, the latin names should be presented in italics. The legends of the graphs should just explain what is presented, not providing comments on the findings.

  • Yes, now the latin names were used in italics.
  • Yes, the legends in the text of graphs were corrected.

How do you explain that in the 2 last images of figure 3, concerning the Trametes versicolor, the mycelia did not totally cover the surface of the petri dish? (since as far as I know, first of all, the culture initially covers all the surface of the glass/petri and then the specimens are being placed). 

  • The growth of T. versicolor on the malt agar soil before staring the mycological test with the beech wood specimens was not always homogenous in all Petri dishes. For starting the mycological test a dominant was grow of fungal mycelium under the plastic mats on which were situated beech specimens.

02.05.2022

Ladislav Reinprecht

Reviewer 2 Report

This research has some reference meaning for wood decay resistance. Here are some suggestions which might be considered about this research:

  1. Though decay resistance is a common problem for wood utilization, it’s more important for wood in outdoor products. Beech wood is usually used as interior furniture, floor, and so on. Is it very important to improve decay resistance of beech wood?
  2. About chemical treatment, line 90-96, either nano-ZnO, nor PEG 6000, they were both used in water solution state when treating beech wood, and the water content in the solution were pretty high. To remove the water, samples were dried after chemical treatment. If the treatment in this research was effective enough to adopt in industry, does it an energy-saving and cost-effective processing method?
  3. One parameter, mass loss, was used to evaluate decay resistance for the Nano-oxide & PEG 6000 & thermally modified wood. How to ensure this parameter objective enough to evaluate all these treatment methods?
  4. Could you give some reason about how these treatment resulting in this decay resistance phenomenon?

Author Response

Answers to Reviewer No 2

This research has some reference meaning for wood decay resistance. Here are some suggestions which might be considered about this research:

  1. Though decay resistance is a common problem for wood utilization, it’s more important for wood in outdoor products. Beech wood is usually used as interior furniture, floor, and so on. Is it very important to improve decay resistance of beech wood?
    • Yes we agree. However, improving of beech wood decay resistance in interiors can be important in some specific situations, e.g., in bathrooms, kitchens, as well as for walls, ceilings or roofs with the possibility of condensation of water vapor.

  1. About chemical treatment, line 90-96, either nano-ZnO, nor PEG 6000, they were both used in water solution state when treating beech wood, and the water content in the solution were pretty high. To remove the water, samples were dried after chemical treatment. If the treatment in this research was effective enough to adopt in industry, does it an energy-saving and cost-effective processing method?
  • Yes, we agree. Energy consumption for wood drying would be high using such modification methods. In this context, better could be a combined chemical-thermal modification of wood realized in one-technological step, e.g., in melt of paraffin + nano-ZnO, or in melt of PEG + nano-ZnO. (NOTE: Such experiments, but without nano-ZnO, were performed in our laboratories by Reinprecht and Repák – published in 2019 and 2022).

  1. One parameter, mass loss, was used to evaluate decay resistance for the Nano-oxide & PEG 6000 & thermally modified wood. How to ensure this parameter objective enough to evaluate all these treatment methods?
  • Yes, but it is OK. The initial weights of specimens before fungal attack m/Before-Fungal-Attack/ depended on their previous modification processes Þ m0, m0/Chemically-Treated/, m0/Thermally-Modified/, or m0/Chemically-Thermally-Modified/) – See the Eq. 2 for mass loss caused by fungi.
  • For example, in the Eq. 2 valid for the thermally modified specimens the initial weights m0/Thermally-Modified/ were used, it means no initial weights m0 valid for the reference specimens or for the specimens before thermal modification.       

  1. Could you give some reason about how these treatment resulting in this decay resistance phenomenon?
  • The summary or even synergistic effect of two modification methods - chemical and thermal - is probably manifested here.

02.05.2022

Ladislav Reinprecht

Reviewer 3 Report

I had the opportunity to review the paper proposed for Forests under the name "Decay Resistance of the Nano-Zinc-Oxide & PEG 6000 & Thermally Modified Wood". The article deals with the very interesting topic which investigate, decay resistance of beech wood which was primary treated with water systems of nano-zinc-oxide (nano-ZnO) and/or polyethylene glycol 6000 (PEG 6000), without/with additional thermal modification at 190 °C/2h.

After reading the article, I have to evaluate positively, but I have few formal mistakes such as:

  • Incorrect stated citation. Lines 54, 73, 121, 169, 199, 230, 236, 244. Correct is [number]. Please correct these lines.
  • Authors stated extensive used references (41) but in discussing the results were used only 7 references so that extensive critical discussion is lacking. Authors have more discussion with other similar articles. Please, add missing discussing.
  • Incorrect stated references. Lines 361, 365. Please correct these lines.

The authors cared about the detailed elaboration of the current state and results. The manuscript has provided interest results such as that was discovered that Nano-ZnO in all combinations – alone, before PEG 6000, before thermal modification – was more effective against the white-rot fungus T. versicolor than against the brown-rot fungus R. placenta etc.

Therefore, after minor editing, I recommend the article for publication in the Forests.

Author Response

Answers to Reviewer No 3

I had the opportunity to review the paper proposed for Forests under the name "Decay Resistance of the Nano-Zinc-Oxide & PEG 6000 & Thermally Modified Wood". The article deals with the very interesting topic which investigate, decay resistance of beech wood which was primary treated with water systems of nano-zinc-oxide (nano-ZnO) and/or polyethylene glycol 6000 (PEG 6000), without/with additional thermal modification at 190 °C/2h.

After reading the article, I have to evaluate positively, but I have few formal mistakes such as:

  • Incorrect stated citation. Lines 54, 73, 121, 169, 199, 230, 236, 244. Correct is [number]. Please correct these lines.

-   By our opinion, in the MDPI rules is valid that in the text of article have to be used number of the reference, but in some cases can be used also the number of the reference together with the name(s) of author(s) of the cited reference.       

  • Authors stated extensive used references (41) but in discussing the results were used only 7 references so that extensive critical discussion is lacking. Authors have more discussion with other similar articles. Please, add missing discussing.
    • Yes, we added some new sentences into the chapter Results and Discussion.

  • Incorrect stated references. Lines 361, 365. Please correct these lines.

-  Yes, these references were corrected.

02.05.2022

Ladislav Reinprecht

Round 2

Reviewer 1 Report

As I have checked the authors have implemented the proposed changes in the revised verion of manuscript towards the improvement of their work. Almost all the changes have been implemented and in my opinion, the manuscript is well-prepared and organized enough to be accepted for publication in this journal. I remain at your disposal for any clarification.

Reviewer 2 Report

According to your reply, since it's better to treat the beech wood by melt nano-ZnO, or melt PEG+nano-ZnO, could you please give some sense for the treatment processing in this research?